# Encipher GAN: An End-to-End Color Image Encryption System Using a Deep Generative Model

Kirtee Panwar [1,†], Akansha Singh [1,*,†], Sonal Kukreja [1,†], Krishna Kant Singh [2,†], Nataliya Shakhovska [3,†] and Andrii Boichuk [3,†]

1    SCSET, Bennett University, Greater Noida 201310, India
2    Deaprtment of CSE, ASET, Amity University, Noida 201303, India
3    Department of Artificial Intelligence, Lviv Polytechnic National University, Lviv 79013, Ukraine
*    Correspondence: akansha1.singh@bennett.edu.in
†    These authors contributed equally to this work.

**Abstract:** Chaos-based image encryption schemes are applied widely for their cryptographic properties. However, chaos and cryptographic relations remain a challenge. The chaotic systems are defined on the set of real numbers and then normalized to a small group of integers in the range 0–255, which affects the security of such cryptosystems. This paper proposes an image encryption system developed using deep learning to realize the secure and efficient transmission of medical images over an insecure network. The non-linearity introduced with deep learning makes the encryption system secure against plaintext attacks. Another limiting factor for applying deep learning in this area is the quality of the recovered image. The application of an appropriate loss function further improves the quality of the recovered image. The loss function employs the structure similarity index metric (SSIM) to train the encryption/decryption network to achieve the desired output. This loss function helped to generate cipher images similar to the target cipher images and recovered images similar to the originals concerning structure, luminance and contrast. The images recovered through the proposed decryption scheme were high-quality, which was further justified by their PSNR values. Security analysis and its results explain that the proposed model provides security against statistical and differential attacks. Comparative analysis justified the robustness of the proposed encryption system.

**Keywords:** deep learning; image encryption; medical images; image reconstruction; structure similarity index metric (SSIM)



## 1. Introduction

With the onset of the Internet of Things era and an extensive increase in digital information transmission over the Internet, security issues such as tampering, personal privacy disclosure and illegal data theft arise from offering convenience to users. Traditional standard techniques such as the DES (Data Encryption Standard) and AES (Advanced Encryption Standard) encrypt textual data and are unsuitable for encrypting digital images. Digital images are crucial information media. As a result, a critical need exists for more secure and effective image encryption techniques. A digital image comprises several pixels with specific numerical values, redundant information and inter-pixel solid correlation. Image encryption is composed of two phases, diffusion and permutation. The pixel positions are exchanged in the permutation phase to eliminate strong correlations among neighboring pixels and hide data. In the diffusion phase, the image pixels are changed and diffused among each other.

Deep learning has tremendous usage in image processing, building chaotic systems, speech recognition, steganography, etc. Recently, deep-learning-based image encryption techniques have received extensive attention. In image encryption techniques, the content remains invisible until decryption using a correct and authorized key. In image cryptography techniques, the original image is encrypted into a cipher image using an encryption

key that recovers the actual image using the decryption key. Cryptanalysis is cracking the information security systems, ciphers, keys, or encrypted images with plaintext attacks and ciphertext-only attacks. Figure 1 shows the relation of cryptanalysis with image encryption systems. Chaotic systems offer good cryptographic properties to image encryption techniques against cryptanalysis attacks, due to their non-linear behavior and sensitivity towards initial conditions.

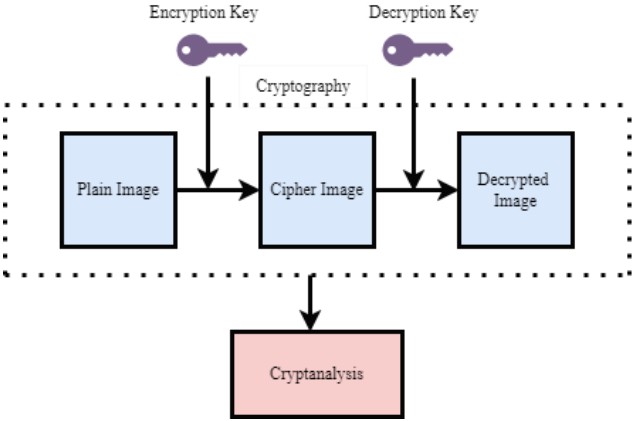

**Figure 1.** Image cryptography and cryptanalysis.

As chaos theory defines itself in a continuum, its application in image cryptography is limited. The range of pixels in the spatial domain of the image is discrete and finite and has a deteriorating effect on the system's security. A proper connection between chaos and cryptography leads to various attacks, despite various suggestions [1] to improve the cryptosystem. There is an immense need for an alternative, secure system for image encryption. Deep learning offers non-linearity to the image encryption system to enhance security against plaintext attacks. Figure 2 shows the image encryption process with a deep model. Deep-learning-based image encryption schemes are gaining much attention from recent researchers, as these schemes achieve equal or better security when compared to traditional encryption schemes. The encryption network of the deep learning network performs the task of encryption, and the decryption network performs the task of decryption. Apart from these two networks, a discriminator network is also employed to confirm that the decryption network regenerates the original image with the maximum possible similarity. This is performed with a cycle-consistent generative adversarial network (Cycle-GAN) [2]. Cycle-GAN shows outstanding performance in image style transfer. In [3], an image encryption/decryption network is designed based on Cycle-GAN: the original images are transformed into cipher images and recovered back through decryption network. The authors suggest its application in the Internet of Medical Things.

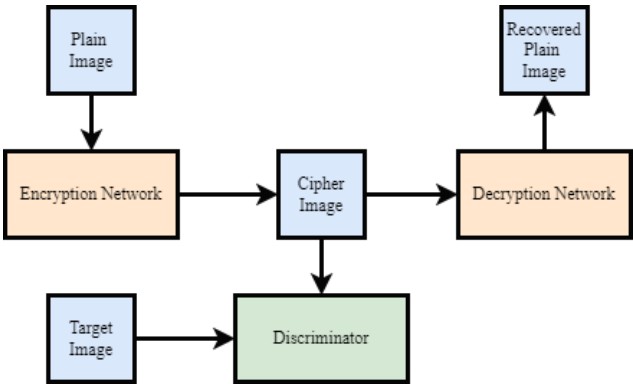

**Figure 2.** Deep-learning-based image encryption.

This paper proposes an image encryption scheme based on the Cycle-GAN network for medical applications. The nonlinearity introduced in the cryptosystem makes it secure against plaintext attacks. Further, the SSIM loss function is used in the proposed network, since medical images require good-quality pictures for better information exchange. The photos recovered through the proposed method can capture full details. The applications of this encryption scheme are limited to medical applications that can afford to lose minute or precise information, such as radiograph images. The images recovered through the proposed method can capture full details. The SSIM loss measures the similarity between pictures, rather than the distance between the pictures. The value of SSIM lies between [0,1], and the loss functions used for the encryption network and discriminator network work together to reflect a measure between real and fake data. The encryption network here can affect the distribution of counterfeit data, which depends on the real data's distribution. This way, the network trains itself. After training, the encryption/decryption network is capable of efficiently encrypting/decrypting an input image. The size of encrypted images is the same as that of the original images. The trainable parameters are the secret keys. The hospital database stores the encrypted images, and only an authorized person with the keys can retrieve the original images.

The remainder of the paper is organized as follows. Section 2 discusses related image encryption schemes in brief. Section 3 illustrates the proposed EncipherGAN, followed by experimental results in Section 4. Section 5 concludes the contribution of the proposed work.

## 2. Related Works

This section discusses conventional research contributions exclusively carried out in the domain of image encryption.

Chen et al. [4] proposed a neural network for impulsive synchronization of the reaction-diffusion mechanism that captures the dynamical behaviors of the system. Further, the system is employed for image encryption applications. Chaotic systems are widely used for cryptography, especially for an image cryptosystem, as it provides security against various traditional attacks, such as plaintext attacks; thus, the neural network system in [4] was also applied in an image cryptosystem. Dridi et al. [5] proposed an image encryption scheme based on combination of chaotic and neural networks. This scheme proved to be more secure and less complex as compared to the existing schemes. Hu et al. [6] proposed image encryption with a stacked auto-encoder network for generating chaotic sequences. The scheme proved efficient due to the stacked autoencoder network's parallel computing abilities and resistance to traditional attacks. Hu et al. [7] proposed novel image steganography without embedding a message into the carrier image using a deep model that improved image security metrics. The scheme shows high extraction phase and resistance against steganalysis algorithms.

Li et al. [8] illustrated an image encryption scheme that generated the encryption key by training a CNN on the CASIA iris dataset [9], extracting the features from iris image and encoding the feature vector using RS error correcting code. This encoded vector was further used to encrypt plain images using XOR operation. Ding et al. [3] improved the previous scheme by training a GAN on Montgomery County's chest X-ray set [10] to generate the encryption key. This scheme showed a larger key space, high resistance to standard image processing attacks, high security due to pseudo randomness and high sensitivity to change. Jin and Kim [11] proposed a DNN-based image encryption scheme that restricted pre-sharing of the keys between systems. It instead created and utilized the keys used in the symmetric key encryption itself, which enhanced security. Another DNN-based robust image encryption scheme was proposed by Maniyath and Thanikaiselvan [12], which was trained on the SIPI image dataset and used chaotic maps to encrypt the image without affecting the image quality. Erkan et al. [13] encrypted the images using a diverse chaotic sequence generated using sensitive keys from training a CNN on the ImageNet database. The initial conditions of the hyperchaotic logistic map used for encryption were determined by the parameters generated through the network.

Fratalocchi et al. [14] developed a two-layer deep neural network to classify silica aerogel (SA) into physical unclonable functions. SA exhibits chaotic behavior, which can be used as keys for cryptographic applications. This system generates a sequence of random key with every possible input condition. Li et al. [15] proposed a Cycle-GAN-based image encryption scheme; the network was trained on a plain-cipher satellite images dataset. Double random phase encoding was used to encrypt the images. Ding et al. [16] proposed another Cycle-GAN based scheme that was trained on a chest X-ray dataset [10]. In addition to the encryption–decryption task, the neural network also identifies the specific object in the cipher image. Bao and Xue [17] investigated the foundations of a strong avalanche effect by studying weaknesses in the previous scheme. A new improved scheme was proposed that also included a diffusion process into Bao et al.'s [18] scheme. The training of the neural network was performed on a satellite images dataset obtained from Google maps to reduce the avalanche affect. The schemed proved to be more efficient but showed low performance in decryption. Networks of Cycle-GANs are extensively employed in the encryption and decryption networks in deep-learning-based image encryption schemes, image steganography [19], etc.

## 3. Proposed Method

The proposed EncipherGAN develops an encrypted image from a given plain image, and vice versa. First, the encryption network $G$ spawns an encrypted image from the plain image, and then decryption network $H$ generates a plain image from the cipher image. The encryption network of Encipher-GAN is trained to generate cipher images that look similar to the target cipher image using encryption network $G$ and discriminator network $D_x$; similarly, the proposed decryption network, $F$, along with the discriminator, $D_y$, is trained with the objective of reconstructing plain images with minimum differences with reference to the original plain image. Figure 3 presents the flow diagram of the proposed encryption method.

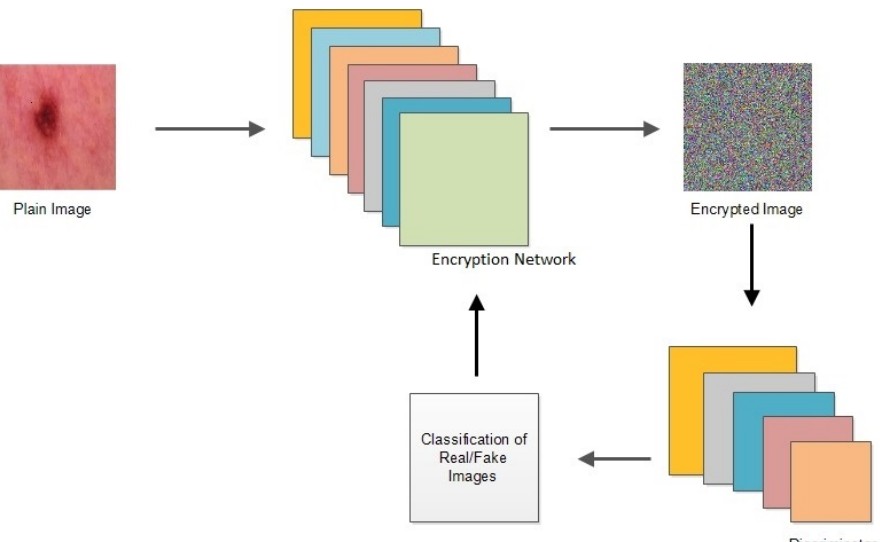

**Figure 3.** Flow diagram of the encryption process.

### 3.1. Encryption Network

The encryption network transforms the style of the plain image (set X) to the style of the cipher image (set Y). The architecture of the encryption network, as illustrated in Figure 4, has three modules, i.e., a feature encoder, a transformation module and a feature decoder. The configuration of the encryption/decryption network, consisting of input layers and a series of convolution layers, is given in Table 1.

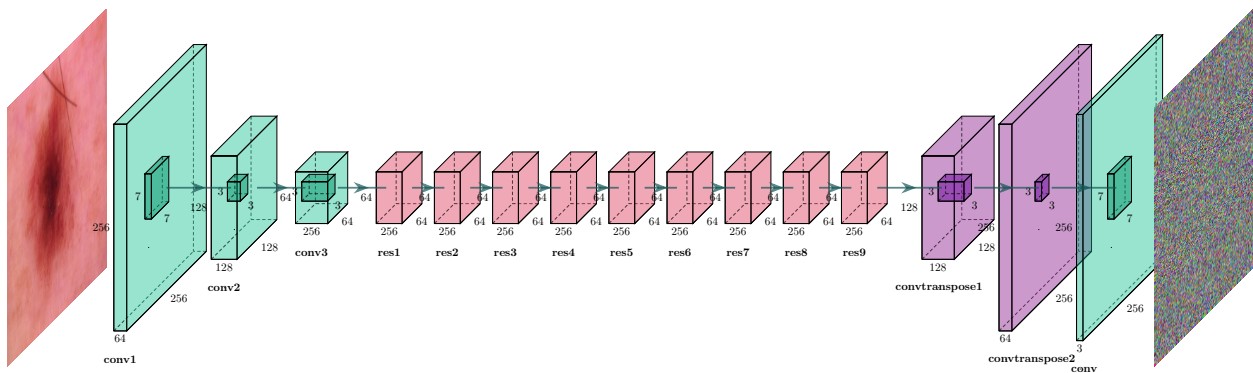

**Figure 4.** Encryption/decryption network.

**Table 1.** Architecture of the encryption/decryption network.

| Layer | Size of Kernel | Normalization Technique | Activation Function | Output | Parameters |
|---|---|---|---|---|---|
| Input | | | | $256 \times 256 \times 3$ | |
| Convolution | $7 \times 7$ | Instance | ReLU | $64 \times 256 \times 256$ | 4704 |
| Convolution | $3 \times 3$ | Instance | ReLU | $128 \times 128 \times 128$ | 18,432 |
| Convolution | $3 \times 3$ | Instance | ReLU | $256 \times 64 \times 64$ | 73,728 |
| Resnet Blocks (9, each with 2 convolutions) | $3 \times 3$ | Batch | ReLU | $256 \times 64 \times 64$ | 2,564,208 |
| Transpose Convolution | $3 \times 3$ | Instance | ReLU | $128 \times 128 \times 128$ | 73,728 |
| Transpose Convolution | $3 \times 3$ | Instance | ReLU | $64 \times 256 \times 256$ | 18,432 |
| Convolution | $7 \times 7$ | Tanh | ReLU | $256 \times 256 \times 3$ | 4704 |

1. Feature Encoder:
   The input image is downsampled with three layers of convolution to extract the features of images.
   The encoder consists of three convolutions, each of which is followed by instance normalization and reLU activation. The first convolution layer has 64 filters of size $7 \times 7$, followed by convolutions with filters of size $3 \times 3$. The encoder downsamples the plain image to extract features.

2. Transformation Module:
   In this phase, the features are transformed by residual blocks. The model is optimized using a ResNet based architecture to enhance the stability of the model. These residual blocks consist of convolution–batch normalization–ReLU–convolution–batch normalization–LReLU. The output of each of these blocks are concatenated and then passed to the decoder. The size of input features and output remains the same during transformation.

3. Feature Decoder:
   The decoder upsamples the transformed features using rounds of transpose convolution layers. In the final convolution layer, these features are mapped to output image of size $256 \times 256 \times 3$.

### 3.2. Discriminator Network

The discriminator network consists of layers of convolution with leaky ReLU activation that extracts features of image, reducing the input volume by a factor of 2. Each convolution is performed with kernel size = 4 and stride = 2. Figure 5 presents the architecture of discriminator network. The output of the discriminator is classification of image similar to target or not; this is achieved with sigmoid activation in the final layer. The configuration of discriminator network is given in Table 2. The discriminator takes in two input images

and compares its features through the network. The network outputs a value which is used to update the encryption network.

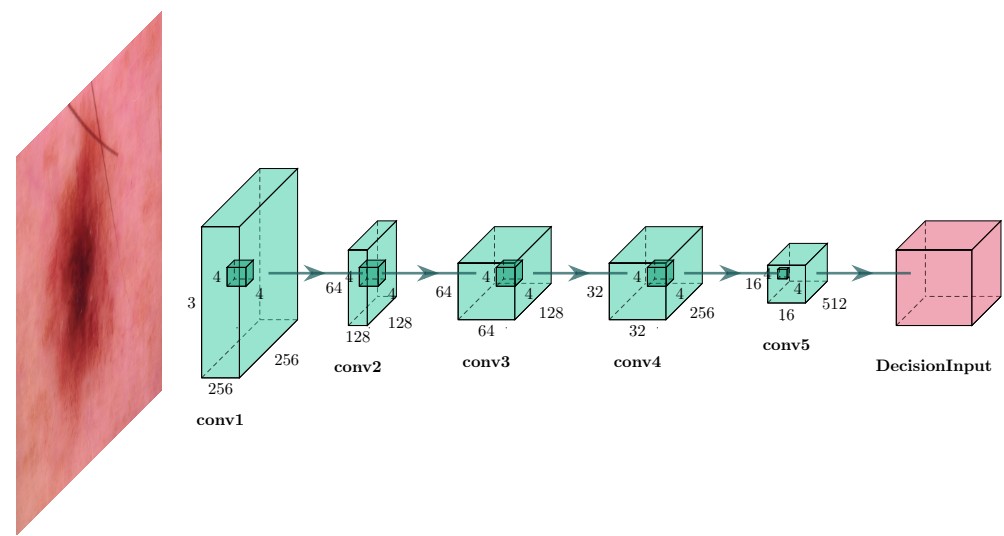

**Figure 5.** Discriminator model.

**Table 2.** Architecture of discriminator.

| Layer | Size of Kernel | Normalization Technique | Activation Function | Output | Parameters |
|---|---|---|---|---|---|
| Input (Two Inputs of same size) | | | | $256 \times 256 \times 3$ | |
| Convolution | $4 \times 4$ | | ReLU | $128 \times 128 \times 64$ | 3136 |
| Convolution | $4 \times 4$ | Instance Normalization | ReLU | $64 \times 64 \times 128$ | 131456 |
| Convolution | $4 \times 4$ | Instance Normalization | ReLU | $32 \times 32 \times 256$ | 524056 |
| Convolution | $4 \times 4$ | Instance Normalization | ReLU | $16 \times 16 \times 512$ | 2098688 |
| Convolution | $4 \times 4$ | Instance Normalization | | $16 \times 16 \times 1$ | 8193 |
| linear | | | Sigmoid | 1 | |

### 3.3. Decryption Network

The decryption network architecture is identical to the encryption network architecture. The input to this network is the cipher image, and the generated image is the plain image.

### 3.4. Training

Discriminator data for training: The discriminator takes data from two sources: the real images (plain/cipher) and the fake ones generated by encryption/decryption network. During training of the discriminator $D_x$, the weights of the encryption network remain the same; it generates cipher images for the discriminator to classify correctly. For training, the discriminator uses one loss function. The adam optimizer is employed to generate a global minimum while updating the weights through backpropagation. The loss function to be minimized for training discriminator is defined as

$$L_D = \mu(SSIM(x,y)) \tag{1}$$

where $\mu = 0.2$ is the hyperparameter to achieve an acceptable equilibrium between the structure aspect of target image and generated image, where SSIM, i.e., the structural similarity index metric, is defined as

$$SSIM(x,y) = \frac{(2u_x u_y + C_1)(\delta_{xy} + C_2)}{(u_x^2 + u_y^2 + C_1)(\delta_x^2 + \delta_y^2 + C_2)} \tag{2}$$

where $C_1 = (k_1 L)^2$; $C_2 = (k_2 L)^2$; $L$ is the maximum value of a pixel; $k_1 = 0.01$ and $k_2 = 0.03$ are constant parameters; $\delta_x$ represents standard deviation of image $x$; and $\delta_{xy}$ represents the covariance of image $x$ and image $y$. The values of $SSIM$ lies in range [0,1], where one indicates completely identical images.

The encryption network learns to create fake data so that the discriminator cannot classify correctly. The encryption network takes input image and generates cipher image. This cipher image is tested for similarity with the real cipher image by the discriminator. If the discriminator gives a value below 0.75, then the weights of the encryption network are updated with the loss function used by encryption network $L_{gen}$, defined as

$$L_{gen}(G) = \mu(1 - SSIM(x,y)) \tag{3}$$

The objective of optimization here is to obtain a global minimum, i.e., making the generated cipher images look similar to the real cipher images. The discriminator output determines whether the weights of the encryption network are to be updated using back-propagation or not. The encryption network is trained with the help of the discriminator. The discriminator's weights remain constant when training of the encryption network is performed. For one epoch, the discriminator is trained, and then the encryption network is trained for one epoch. This cycle is repeated continuously for training of the entire network.

The decryption network is similar to the encryption network and transforms the generated cipher image back to a plain image. The training of discriminator network is performed in a similar way with the help of discriminator $D_y$. The encryption system transforms image $x$ from domain $X$ into a cipher image through mapping $G(x)$, and the decryption system must be able to retrieve the plain image $x$ back from cipher image $G(x)$—i.e., $x \rightarrow G(x) \rightarrow F(G(x)) \approx x$; therefore, forward-cycle consistency loss [2] is defined as

$$L_{cyc} = \mathbb{E}_{x \sim P_{data(x)}}[||F(G(x)) - x]] + \mathbb{E}_{y \sim P_{data(y)}}[(G(F(y)) - y||_1], \tag{4}$$

and is also employed during training of encryption/decryption network. This loss penalizes the output image $\hat{x}$ generated by the decryption network when it deviates in content from the target $x$ and vice versa.

Total Loss of the Network

Total loss of the encryption network is defined as

$$L(G,F) = L_{gen}(G) + L_D + \lambda L_{cyc} \tag{5}$$

The value of constant $\lambda = 10$.

Finally, the information in the plain image is converted into a cipher image with perceptual properties of the real cipher image. The parameters in encryption/decryption network are the final parameters that form the private key. The plain image is regenerated from the cipher image with cycle consistency loss. Mean-square error (MSE) calculates the pixel-level differences between two images that do not account for the adjacent pixel correlation between images. In this paper, the structural features of two images are considered with the SSIM index while selecting the loss function, which is an essential metric for measuring image quality.

### 3.5. Encryption/Decryption Algorithm

The encryption is performed efficiently with the encryption network once the trained network has been obtained. During training, the parameters are randomly initialized, and the parameters of the network obtained after training are the secret keys of the encryption system. Using these keys, the plain image can be encrypted, and from the encrypted image, the plain image can be retrieved. The plain image to be encrypted must belong to a class of images on which the network is trained. The proposed model was trained on skin cancer images taken from [20]. The encryption algorithm is given below (Algorithm 1):

---

**Algorithm 1** Encryption model.

---

**Input:** Plain Image $I$ of size $W \times H \times 3$ and Secret Key ($K1$)
**Output:** Cipher Image, $C$

　*Initialisation* : Assign trained weights to the encryption network using secret key $K1$.

1: Data preprocessing : The input image has pixels in range from 0 to 255 which is normalized using normalization technique.
2: The trained network takes in normalized input image and through a series of convolution and activation functions as defined in Table 1, the network generates cipher image of same size as the input.

---

## 4. Security Analysis

The nonlinearity of the model enhances the security of the encryption scheme as compared to chaotic encryption schemes, which are vulnerable to phase-space-reconstruction attacks [21]. The depth of the encryption/decryption network is 15, and the number of parameters in the network is approximately 2,800,000. These parameters obtained after training the network are used as the secret keys for encryption/decryption. Due to depth of the deep learning model, the complexity for cryptanalytic attacks is further increased. Various factors that affect security the encryption system are discussed in this section.

### 4.1. Secret Key Space

The size of the key space determines its potential to counter brute force attacks. The trained parameters of each layer contribute to the key space. Due to the depth of the encryption network, the key space of the encryption system is large, i.e., $2^{Parameters*32}$. Due to the large key space, a brute force attack is not a practical approach for attackers.

### 4.2. Secret Key Sensitivity

Key sensitivity can be determined by performing encryption with different keys. Each time the encryption network is trained, a different set of parameters is obtained. Therefore, by training the network two times, two different sets of parameters are obtained. The same image was used to perform encryption with each of the trained encryption networks, and as a result, two different cipher images were obtained, as shown in Figure 6b,c. The PSNR value of between cipher image shown in Figure 6b,c is 10.54, which shows the high sensitivity of the keys used in the encryption system.

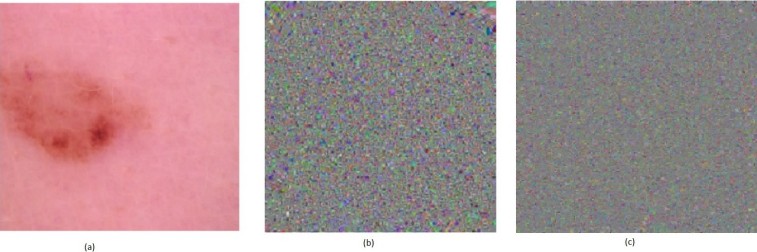

**Figure 6.** Plain image. (**a**) Image 1. (**b**) Cipher Image 1; (**c**) Cipher Image 2—generated with different sets of keys.

### 4.3. Analysis of the Histogram of Generated Images

Analysis of the histogram of plain images, as shown in Figure 7a–c, and the cipher images shown in Figure 7d–f, was performed. The histogram distribution of encrypted images determines its resistance towards statistical attacks. The histogram of cipher images that have been generated through proposed encryption system is different from the histogram of plain images, though they closely resemble each other. They have the same distributions as their plain images. The similarity in the histograms of encrypted images is shown in Figure 8. This similarity between histograms makes it difficult to obtain any information regarding plain images and is suitable for encrypting medical images.

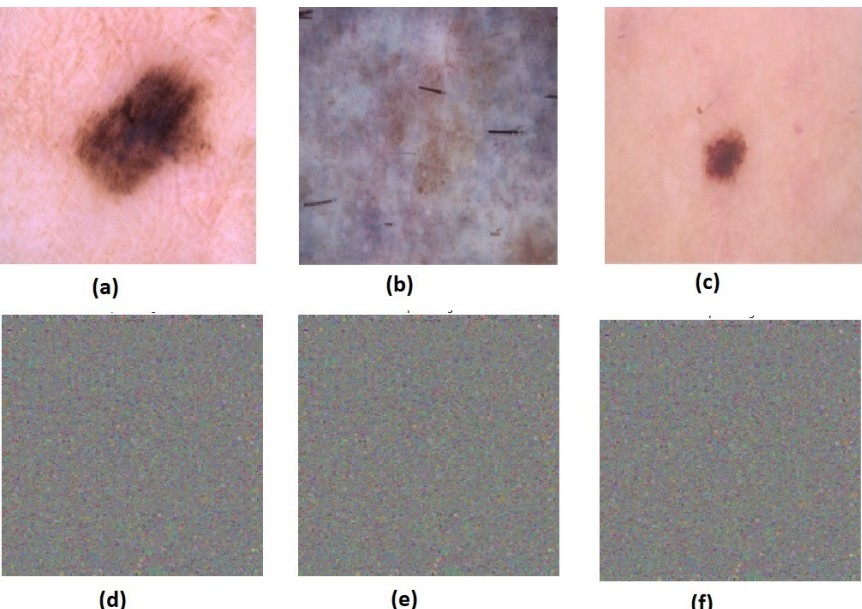

**Figure 7.** Plain image. (**a**) Image 1, (**b**) Image 2, (**c**) Image 3. (**d**) Cipher Image 1. (**e**) Cipher Image 2. (**f**) Cipher Image 3.

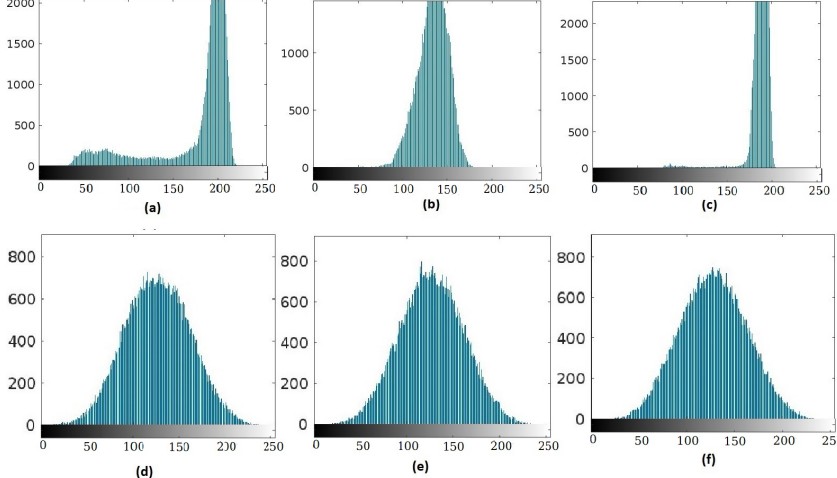

**Figure 8.** Histogram of plain images (**a–c**) and corresponding cipher images (**d–f**).

### 4.4. Image Information Entropy

Image information entropy determines the uncertainty of pixels in the cipher image and is defined as

$$IE = -\sum_{i=0}^{255} p_i \log p_i \qquad (6)$$

where $p_i$ is the probability of pixel $i$ in an image. An image with ideally random pixels has image information entropy value eight. Table 3 displays the entropy values for plain image pixels and their cipher images' pixels. It is apparent from Table 3 that the entropy values in the cipher images obtained from the image encryption scheme are higher than those of the plain images.

**Table 3.** Image information entropy values of plain images and cipher images.

| Images | Image 1 | Image 2 | Image 3 |
|--------|---------|---------|---------|
| Plain | 7.15 | 6.27 | 6.04 |
| Cipher | 7.40 | 7.36 | 7.38 |

*4.5. Correlation Analysis*

Neighboring pixel correlation determines the strength of an encryption model against statistical attacks. Adjacent pixel correlation in the horizontal direction was calculated by randomly choosing 2000 horizontally adjacent pixels and then calculating the correlation coefficient between each of the adjacent pixels using

$$corr = \frac{N \sum_{i=1}^{N} (x_i y_i) - \sum_{i=1}^{N} x_i \times \sum y_i}{(N \sum_{i=1}^{N} x_i^2 - (\sum_{i=1}^{N} x_i)^2) \times (N \sum_{i=1}^{N} y_i^2 - (\sum_{i=1}^{N} y_i)^2)} \tag{7}$$

Similarly, the correlation coefficient was calculated for vertical and diagonal directions. Table 4 displays the adjacent pixel correlation coefficients among pixels in plain images and cipher images.

It is apparent in Table 4 that the adjacent pixel correlation is low in the cipher images compared to their plain images. Figure 9a–c show the dispersal of pixels in three directions (horizontal, vertical and diagonal directions) in plain images. Pixel distribution in its corresponding cipher images, presented in Figure 9d–f for horizontal, vertical and diagonal directions, justifies that the correlation among neighboring pixels in the plain image is minimized successfully in its cipher image.

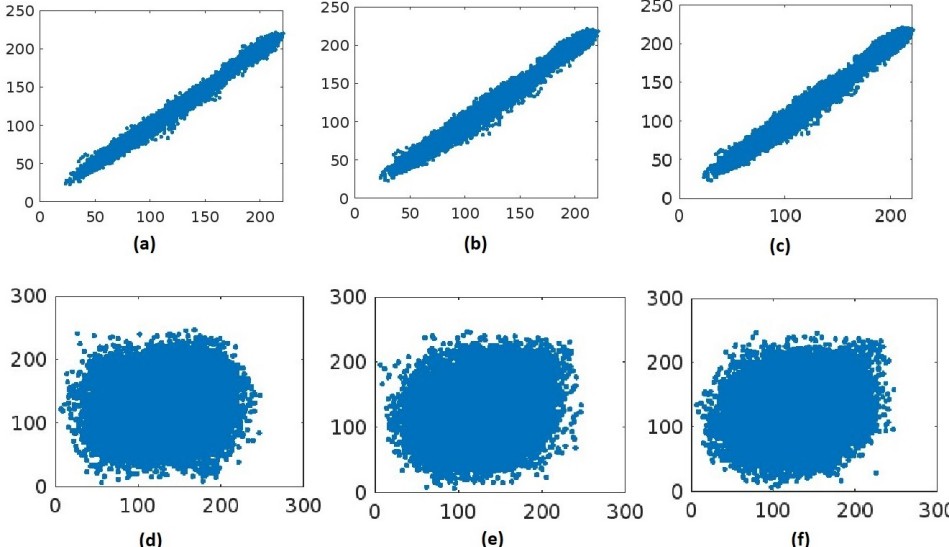

**Figure 9.** Adjacent pixel correlations (horizontal, vertical and diagonal directions) of plain images (**a**–**c**) and corresponding cipher images (**d**–**f**).

**Table 4.** Correlation coefficient values among adjacent pixels.

| Images | Horizontal | Vertical | Diagonal |
|---|---|---|---|
| Image 1 | 0.9976 | 0.9986 | 0.9959 |
| Cipher | 0.4812 | 0.4584 | 0.2169 |
| Image 2 | 0.9976 | 0.9984 | 0.9963 |
| Cipher | 0.5090 | 0.4538 | 0.2043 |
| Image 3 | 0.9977 | 0.9986 | 0.9961 |
| Cipher | 0.5203 | 0.4147 | 0.1782 |

## 5. Performance Analysis

The proposed model was tested on two sets of data, one that has the plain images to be encrypted and another that has the cipher images, which does not reveal any information regarding the plain image. These two sets of data were obtained from a skin cancer dataset in [20], from which plain images were taken, and encrypted images were used that were obtained by encrypting these plain images with the image encryption algorithm proposed in [22]. All input images and generated images are of size $256 \times 256 \times 3$. After training the encryption–decryption model using a skin cancer dataset, the proposed encryption–decryption system is capable of encrypting/decrypting random plain text/cipher text, respectively.

### 5.1. Optimization Process

The models are trained with the adam version of stochastic gradient descent (SGD) with a batch size of one, which denotes that after each image was generated, weights of the network were updated. Learning rate 0.0002, $beta_1 = 0.5$ and $beta_2 = 0.999$ were employed for the adam optimizer. The network's performance was optimized as the training proceeded. This is apparent in the quality of encrypted images obtained at different intervals during training. Figure 10 shows the improvement in cipher image quality as more epochs were completed. Figure 10b shows an encrypted image after one epoch, Figure 10c after 10 epochs and Figure 10d after 40 epochs, which reflects the continuous optimization process of the model. Finally, after 100 epochs, as shown in Figure 10e, a good quality cipher image was obtained.

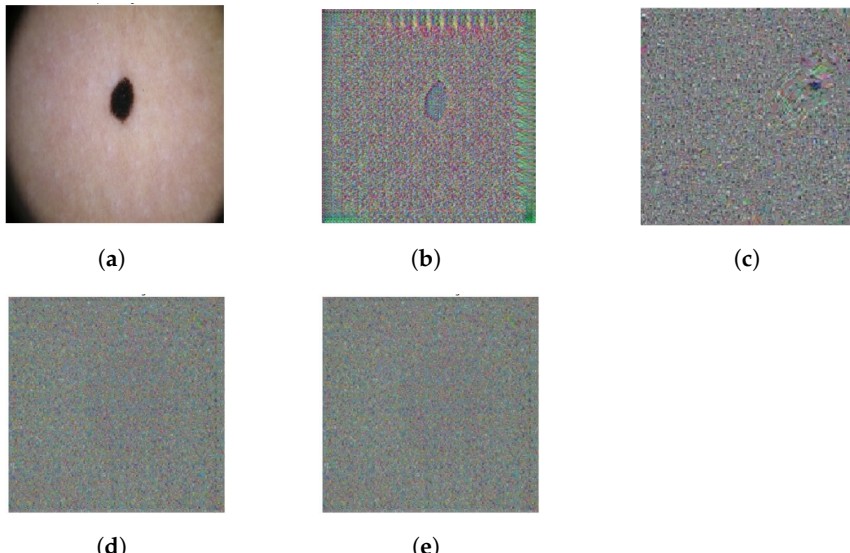

(a)  (b)  (c)

(d)  (e)

**Figure 10.** Comparison of the quality of encrypted images as training proceeded: (**a**) Original Image; (**b**) Cipher Image (epoch 1); (**c**) Cipher Image (epoch 10); (**d**) Cipher Image (epoch 40); (**e**) Cipher Image (epoch 100).

### 5.2. Quality of Recovered Image

The natural image has a high correlation among pixels, along with potential structural features. The proposed model employs the SSIM index as a loss function, which captures the essential structure of images in the generated (cipher image) and recovered (original image) images. After the model's training, the proposed network encrypts original images, as shown in Figure 11, to obtain encrypted images. The proposed decryption network generates recovered images, as shown in Figure 11. The similarity between recovered images and original plain images' mosaics justifies the recovered images' quality. Peak signal-to-noise ratio was calculated amongst the original image and the generated image obtained through the proposed decryption network. The values of PSNR given in Table 5 justify the network's performance. The proposed decryption network recovered good-quality images. Table 5 compares PSNR values obtained with the proposed network to PSNR values obtained with other encryption networks. The value of PSNR for the proposed network is higher than those reported in [3,23].

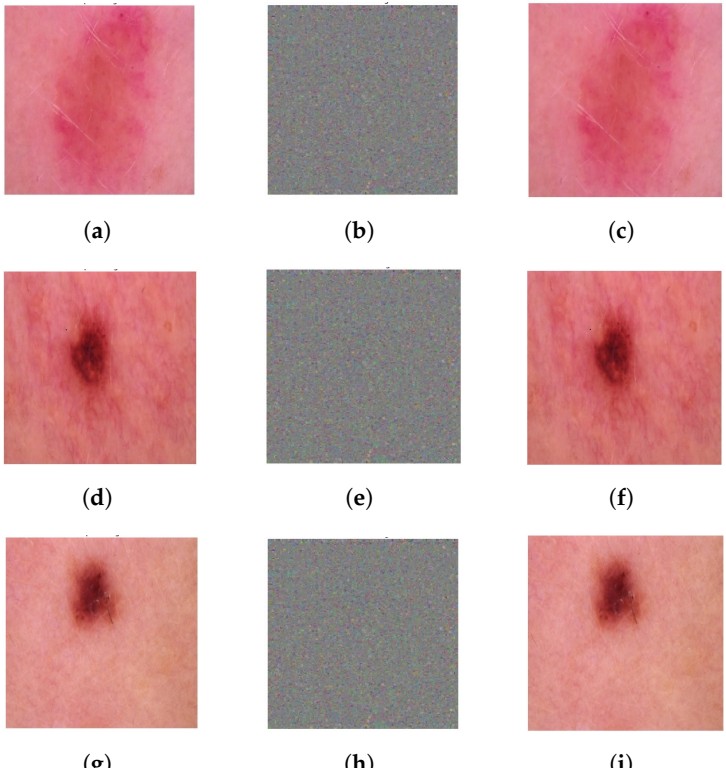

**Figure 11.** Quality of recovered images generated with the decryption network: (**a**) Original Image; (**b**) Encrypted Image of (**a**); (**c**) Recovered Image from (**b**); (**d**) Original Image; (**e**) Encrypted Image of (**d**); (**f**) Recovered Image from (**e**); (**g**) Original Image; (**h**) Encrypted Image of (**g**); (**i**) Recovered Image from (**h**); (Column 1 displays original images. Column 2 displays encrypted images generated with the encryption network. Column 3 shows recovered images generated with the decryption network.)

**Table 5.** PSNR values between plain images and recovered plain images.

| Methods | Average PSNR |
|---|---|
| Proposed Method | 39.9703 |
| DeepEDN [3] | 36.514 |
| EncryptGAN [23] | 17.5992 |
| Image encryption system with CNN denoiser [24] | 24.8975 |
| Optical Image Encryption using Deep Learning [25] | 30.0000 |

### 5.3. Comparative Analysis

Chaos-based image encryption schemes generate good-quality cipher images with high randomness among pixels of the cipher images [12,13], as observed in the entropy values and correlation coefficient values in Table 6. However, various cryptanalytic works [22,26] have shown the vulnerability of chaos-based image encryption schemes to plaintext attacks [27]. The chaotic systems are defined on a set of real numbers and then normalized to a small group of integers in the range 0–255; this affects the security of such cryptosystems [27–29]. The proposed encryption scheme resists plaintext attacks due to non-linearity introduced through the deep convolutional neural network. A detailed comparative analysis in Table 6 further justifies the ability of the proposed system to provide robustness compared to existing methods through high PSNR values of the proposed encryption system as compared to those of [3,24,25].

**Table 6.** Comparative analysis of image encryption schemes with the proposed encryption system.

| Ref. | Technique | PSNR between Original Image and Recovered Image | SSIM between Original Image and Recovered Image | Correlation Coefficient | Image Entropy of Cipher Image |
|---|---|---|---|---|---|
| Proposed | Deep learning based encryption | **39.9703** | **0.9972** | 0.3855 | 7.36 |
| [12] | Deep learning based secret keys and chaos-based encryption | inf | 1 | 0.0149 | 7.98 |
| [13] | Deep learning based secret keys and chaos-based encryption | inf | 1 | 0.00002 | 7.99 |
| [3] | Deep learning based image encryption scheme | 36.514 | 0.90000 | – | 7.95 |
| [24] | Optical image encryption scheme using deep convolutional neural network | 24.8975 | 0.8885 | – | – |
| [25] | Optical Image encryption and Hiding using deep learning | 30.0000 | 0.9306 | – | – |

### 6. Conclusions

In this paper, an efficient and secure end-to-end encryption/decryption network is proposed based on deep learning processes. Two networks were designed for encryption and decryption using a skin cancer dataset. The encryption and decryption networks are trained simultaneously to generate images with the structural feature extraction capability of a CNN. The loss function, SSIM, is employed for training these networks, which also takes into account the structure, luminance and contrast of the target image. The cryptographic properties of its cipher images are comparable to those of cipher images generated by traditional image encryption systems. The proposed decryption system is capable of retrieving robust plain images with good PSNR values. The trainable parameters of the network are the keys of the encryption–decryption system. The encryption system obtained after training has a large number of parameters; therefore, the encryption system has a large key space. The security of the proposed encryption system is compared with that

of other state-of-art methods based on deep learning processes, and it is apparent that proposed encryption/decryption system is efficient, secure and robust.

**Author Contributions:** The contribution of authors are: Conceptualization, A.S. and K.K.S.; methodology, A.S., K.P., N.S. and A.B.; software, K.P.; validation, K.P., N.S. and A.B.; formal analysis, S.K. and K.K.S.; writing—original draft preparation, S.K. and K.P.; writing—review and editing, A.S. and K.K.S. All authors have read and agreed to the published version of the manuscript.

**Funding:** This research received no external funding.

**Data Availability Statement:** Not applicable.

**Conflicts of Interest:** The authors declare no conflict of interest.

## Abbreviations

The following abbreviations are used in this manuscript:

| DES | Data encryption standard |
|---|---|
| AES | Advanced encryption standard |
| Cycle GAN | Cycle generative adversarial network |
| SSIM | Structural similarity index |
| MSE | Mean-squared error |
| PSNR | peak signal-to-noise ratio |
| CNN | Convolutional neural network |
| DNN | Deep neural network |
| SA | Silica aerogel |
| IE | Image entropy |
| DeepEDN | Deep learning based image encryption and decryption Network |

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
