# Peer review of "Encipher GAN: An End-to-End Color Image Encryption System Using a Deep Generative Model"

_systems, doi:10.3390/systems11010036_

Round 1
Reviewer 1 Report
In this paper, an encryption system is proposed based on deep learning approach. The proposed model shows good encryption properties and decryption network shows good retrieval properties. However, the manuscript needs to be revised before it can be accepted. The following modifications are suggested before acceptance:
1 The authors need to better explain the context of this research, including why the research problem is important.
2. The introduction should clearly explain the key limitations of prior work that are relevant to this paper.
3. All the equations need to be written as per format instructions.
4. In Fig. 3, the second block is Generator but in section 3.1 there is no explanation about Generator. Is Encryption network same as Generator? The authors must clearly explain about the same.
5. Correct the ordering of Table 1. and Table 2.
6. The contributions must be clearly stated in the abstract.
7. The manuscript needs to be revised for grammatical error corrections. Overall typos and formatting must be checked.
8. The contents in Table 1 and Table 2 can be described well with a text sentence.
Author Response
In this paper, an encryption system is proposed based on deep learning approach. The proposed model shows good encryption properties and decryption network shows good retrieval properties. However, the manuscript needs to be revised before it can be accepted. The following modifications are suggested before acceptance:
- The authors need to better explain the context of this research, including why the research problem is important.
Answer: The abstract and introduction has been revised explaining the context of this research and its relevance.
- The introduction should clearly explain the key limitations of prior work that are relevant to this paper.
Answer: The introduction has been revised explaining key limitations of prior work.
- All the equations need to be written as per format instructions.
Answer: The equations have been written as per latex format.
- In Fig. 3, the second block is Generator but in section 3.1 there is no explanation about Generator. Is Encryption network same as Generator? The authors must clearly explain about the same.
Answer: The correction in Fig 3. Has been made, it is encryption network.
- Correct the ordering of Table 1. and Table 2.
Answer: The ordering is corrected
- The contributions must be clearly stated in the abstract.
Answer: The abstract has been revised.
- The manuscript needs to be revised for grammatical error corrections. Overall typos and formatting must be checked.
Answer: Corrections have been made, the manuscript has been revised.
- The contents in Table 1 and Table 2 can be described well with a text sentence.
Answer: Description of Table 1 and Table 2 has been added within text.
Reviewer 2 Report
This paper proposes an image encryption system developed using deep learning to realize the secure transmission of medical images over an insecure network. There are some questions in the present work, and the author should answer them before publication.
1.In the information age, we need to transmit large-scale picture data. Will this encryption method lead to the increase of transmission cost and storage cost?
2.In the medical field, the slight difference of image recovery quality will lead to the error of information transmission. Is the image recovered by this method applicable to the medical field?
3.The clarity of the pictures in this article needs to be improved, and some of the details are blurred.
4.Is there any comparison of the security performance of the different methods?
Author Response
This paper proposes an image encryption system developed using deep learning to realize the secure transmission of medical images over an insecure network. There are some questions in the present work, and the author should answer them before publication.
1.In the information age, we need to transmit large-scale picture data. Will this encryption method lead to the increase of transmission cost and storage cost?
Answer: This method will not increase transmission and storage costs because the idea is to replace the original images with encrypted images. The only extra information that needs to be stored is secret keys. The revised introduction explains the same.
2.In the medical field, the slight difference of image recovery quality will lead to the error of information transmission. Is the image recovered by this method applicable to the medical field?
Answer: It is suitable for medical imaging applications where sharp or precise information can be ignored; explanation has been added in introduction.
3.The clarity of the pictures in this article needs to be improved, and some of the details are blurred.
Answer: The figures have been updated.
4.Is there any comparison of the security performance of the different methods?
Answer: A section has been added for comparative analysis (sec 5.3)
Reviewer 3 Report
The paper proposes an image encryption system developed using deep learning to realize the secure transmission of medical images over an insecure network. The loss function employs the Structure similarity index metric (SSIM) to train the encryption/decryption network to achieve the desired output. Recovered images obtained through the proposed decryption scheme are high-quality, further justified by PSNR values.
After reviewing the paper, comments are given as follows:
1. The logic of the abstract is not strict. The problems to be solved in the paper and the innovation need to be explained clearly.
2. The Figures in the paper are blurry. It is suggested that the author redraw them, especially the text in Figure 4. Some illustrations in Figure 10 and Figure 11 are blocked.
3. The font size in the Table 3, 4, and 5 should not be larger than the text.
4. In Security Analysis part, comparative experiments with reference 12, 13, 23, and 24 are needed to show the advantages and effectiveness of the methods in the paper.
5. There are too few references. The author is suggested to supplement relevant references.
Author Response
The paper proposes an image encryption system developed using deep learning to realize the secure transmission of medical images over an insecure network. The loss function employs the Structure similarity index metric (SSIM) to train the encryption/decryption network to achieve the desired output. Recovered images obtained through the proposed decryption scheme are high-quality, further justified by PSNR values.
After reviewing the paper, comments are given as follows:
- The logic of the abstract is not strict. The problems to be solved in the paper and the innovation need to be explained clearly.
Answer: The abstract has been revised, problems solved in this paper has been highlighted clearly.
- The Figures in the paper are blurry. It is suggested that the author redraw them, especially the text in Figure 4.Some illustrations in Figure 10 and Figure 11 are blocked.
Answer: Figures have been updated. Fig. 4 have been redrawn. Corrections have been made for Fig. 10 and Fig. 11.
- The font size in the Table 3, 4, and 5 should not be larger than the text.
Answer: The font size has been corrected.
- In Security Analysis part, comparative experiments with reference 12, 13, 23, and 24 are needed to show the advantages and effectiveness of the methods in the paper.
Answer: A section for comparative analysis has been added (sec 5.3)
- There are too few references. The author is suggested to supplement relevant references.
Answer: Relevant references have been added.
Round 2
Reviewer 3 Report
The authors have modified paper according to the reviewers' comments and the quality has been improved significantly.
The resolution of Figure 8 and Figure 9 should be still improved, so minor revision is suggested.
Author Response
The resolution of fig. 8 and fig. 9 is improved in the revised manuscript.